# Fair Bayes-Optimal Classifiers Under Predictive Parity

**Xianli Zeng**
NUS (Chongqing) Research Institute
Chongqing, China
zengxl19911214@gmail.com

**Edgar Dobriban**[*]
University of Pennsylvania
Philadelphia, PA 19104
dobriban@wharton.upenn.edu

**Guang Cheng**[†]
University of California, Los Angeles
Los Angeles, CA 90095
guangcheng@ucla.edu

## Abstract

Increasing concerns about disparate effects of AI have motivated a great deal of work on fair machine learning. Existing works mainly focus on independence and separation-based measures (e.g., demographic parity, equality of opportunity, equalized odds), while sufficiency-based measures such as predictive parity are much less studied. This paper considers predictive parity, which requires the same probability of success given a positive prediction, among different protected groups. We prove that, if the overall performances of different groups vary only moderately, all fair Bayes-optimal classifiers under predictive parity are group-wise thresholding rules. Perhaps surprisingly, this may not hold if group performance levels vary widely; in which case we find that predictive parity among protected groups may lead to within-group unfairness. We then propose an algorithm we call FairBayes-DPP, aiming to ensure predictive parity when our condition is satisfied. FairBayes-DPP is an adaptive thresholding algorithm that aims to achieve predictive parity, while also seeking to maximize test accuracy. We provide supporting experiments conducted on synthetic and empirical data.

## 1 Introduction

Due to the increasing ability to handle massive data with extraordinary model accuracy, machine learning (ML) algorithms have achieved remarkable success in many applications, such as computer vision [44, 46, 23, 47] and natural language processing [45, 49, 12, 55]. However, empirical studies have also revealed that ML algorithms may incorporate bias from the training data into model predictions. Due to historical biases, vulnerable groups are often under-represented in available data [27, 48]. Moreover, risk minimization may further inadvertently introduce biases that are not in the data [60, 24]. As a consequence, without fairness considerations, ML algorithms can be systematically biased against certain groups defined by protected attributes such as race and gender.

As algorithmic decision-making systems are now widely integrated in high-stakes decision- making processes, such as in healthcare [21] and criminal prediction [27], fair machine learning has grown rapidly over the last few years into a key area of trustworthy AI. A main task in fair machine learning is to design efficient algorithms satisfying fairness constraints with a small sacrifice in model accuracy. This field has made substantial progress in recent years, as many effective approaches have been proposed to mitigate algorithmic bias [57, 33, 3, 52, 4, 9, 35, 10, 25, 5, 58].

---

[*]https://statistics.wharton.upenn.edu/profile/dobriban/
[†]http://www.stat.ucla.edu/~guangcheng/

36th Conference on Neural Information Processing Systems (NeurIPS 2022).

An important fundamental benchmark for fair classification is provided by fair Bayes-optimal classifiers, which maximize accuracy subject to fairness [36, 58]. A key class of classifiers is group-wise thresholding rules (GWTRs) over the feature-conditional probabilities of the target label, for each protected group (e.g., probability of repaying a loan given income). Intuitively, being a GWTR is a minimal requirement for within-group fairness: the most qualified individuals are selected in every group. [8, 36, 7, 1, 43, 58] have studied fair Bayes-optimal classifiers under various fairness constraints and proved that, for many fairness metrics, the optimal fair classifiers are GWTRs. Moreover, the associated thresholds can be learned efficiently [36, 58].

Current literature on Bayes-optimality focuses mainly on the independence- and separation-based fairness measures (e.g., demographic parity, equality of opportunity, equalized odds; see Section 2.1 for definitions and a review). However, the theoretical benchmark for some sufficiency-based measures such as predictive parity are not well understood, possibly due to the complexity of their constraints. Liu et al. [31] show that a particular sufficiency-based measure, group calibration, is implicitly favored by unconstrained optimization: calibration error is bounded by the excess risk over the unconstrained Bayes-optimal classifier. Hebert-Johnson et al. [24] proposed a multicalibration method that guarantees calibrated predictions for several subpopulations. For selective classification, Lee et al. [30] find that sufficiency-based representation learning leads to fairness. In this paper, we consider predictive parity, which requires that the positive predictive value (probability of a successful outcome given a positive prediction) be similar among protected groups. In credit lending, for example, predictive parity requires that, for individuals who receive the loans, the repayment rates in different protected groups are the same. Although predictive parity is often applied to assess an algorithm in recidivism prediction [19, 13, 6], little is known about (1) what are the optimal fair classifiers under predictive parity and (2) how to learn them effectively. In this paper, we aim to answer these two questions. We first study fair Bayes-optimal classifiers under predictive parity. Perhaps surprisingly, our theoretical results reveal that the optimal fair classifiers may or may not be a GWTR, depending on the data distribution. We identify a sufficient condition under which all fair Bayes-optimal classifier are GWTRs. Without this condition, we show that fair Bayes-optimal classifiers may not be a GWTR when the minority group is more qualified than the majority group. In these cases, predictive parity may have limitations as a fairness measure, as it can either lead to within-group unfairness for the minority group or results in accuracy loss. Our findings are a reminder that the improper use of fairness measures may result in severe unintended consequences. Careful analysis before applying fairness measures is necessary.

We then develop an algorithm, FairBayes-DPP, aiming for predictive parity. Our method is a two-stage plug-in method. In the first step, we use standard learning algorithms to estimate group-wise conditional probabilities of the labels. In the second step, we first check our sufficient condition, and then apply a plug-in method for estimating the optimal thresholds under fairness for each protected group.

We summarize our contributions as follows.

- We show that Bayes-optimal classifiers satisfying predictive parity may or may not be group-wise thresholding rules (GWTRs), depending on the data distribution.
- We identify a sufficient condition under which all fair Bayes-optimal classifiers are GWTRs. However, when the sufficient condition is not satisfied, the fair Bayes-optimal classifier may lead to within-group unfairness for the minority group.
- We propose the FairBayes-DPP algorithm for binary fair classification. The proposed FairBayes-DPP is computationally efficient, showing a solid performance in our experiments.

## 2 Related Literature

### 2.1 Fairness Measures

Various fairness metrics have been proposed to measure aspects of disparity in ML. Group fairness [2, 15, 22] targets statistical parity across protected groups, while individual fairness [26, 29, 41] aims to provide nondiscriminatory predictions for similar individuals. In general, group fairness measures can be categorized into three categories.

The first group consists of independence-based measures, which require independence between predictions and protected attributes; this includes demographic parity [28, 57] and conditional

statistical parity [8, 1]. In credit lending, independence means that the proportion of approved candidates is the same across different protected groups. However, as discussed in [22], independence-based measures have limitations; and applying them often leads to a substantial loss of accuracy.

The second group consists of separation-based measures, which require conditional independence between predictions and protected attributes, given label information. Typical examples in this group are equality of opportunity [22, 59] and equalized odds [22, 56]. In credit lending, separation-based measures require, that the individuals who will pay back (or default on) their loan have an equal probability of getting the loan, despite their race or gender. Compared to independence-based measures, separation-based measures take label information into account, allowing for perfect predictions that equal the label. However, these measures are hard to validate in certain applications as the label information is often unknown for some groups. For example, the repayment status is missing for individuals whose loan application is declined.

As a result, measuring predictive bias is more widely applicable. This leads to the third class, sufficiency-based measures [39, 6, 31], where the label is required to be conditionally independent of the protected attributes, given the prediction. In credit lending, this requires that among the approved applications, the proportion of individuals who pay back the loan is equal across different groups. Unlike independence- and separation-based measures that are well studied with solid theoretical benchmarks and efficient algorithms, some sufficiency-based measures, such as predictive parity, are less commonly investigated. A possible reason is that conditioning on the prediction leads to a complex constraint, which is thus challenging to study and enforce algorithmically.

## 2.2 Algorithms Aimed at Fairness

Literature on algorithms for fairness has grown explosively over the past decade. Existing algorithms for fairness can be categorized broadly into three categories. The first category is pre-processing algorithms aiming to remove biases from the training data. Examples include transformations [17, 34, 3, 25], fair representation learning [57, 33, 35, 10] and fair data generation [53, 42, 54, 40]. The second group is in-processing algorithms, which handle fairness constraints during the training process. Two common strategies are penalized optimization [20, 38, 9, 5] and adversarial training [59, 50, 52, 4]. The former incorporates fairness measures as a regularization term into the optimization objective and the latter tries to minimize the predictive ability of the model with respect to the protected attribute.

The third group is post-processing algorithms, aiming to remove disparities from the model output. The most common post-processing algorithm is the thresholding method [18, 36, 1, 43, 58], adjusting thresholds for every protected group to achieve fairness. In this paper, we propose a post-processing algorithm, FairBayes-DPP, to estimate the fair Bayes-optimal classifier under predictive parity.

## 3 Problem Formulation and Notations

In this paper, we consider classification problems where two types of feature are observed: the usual feature $X \in \mathcal{X}$, and the protected feature $A \in \mathcal{A}$. For example, in loan applications, $X$ may refer to common features such as education level and income, and $A$ may correspond to the race or gender of a candidate. As multiclass protected attributes are often encountered in practice, we allow $\mathcal{A}$ to have any number $|\mathcal{A}| \geq 1$ of classes, and let $\mathcal{A} = \{1, 2, ..., |\mathcal{A}|\}$. We denote by $Y$ the ground truth label. In credit lending, $Y$ may correspond to the status of repayment or defaulting on a loan. The output $\hat{Y}$ of the classifier aims to predict $Y$ based on observed features. We consider randomized classifiers defined as follows:

**Definition 3.1** (Randomized classifier). *A randomized classifier is a measurable function[3] $f :$ $\mathcal{X} \times \mathcal{A} \to [0, 1]$, indicating the probability of predicting $\widehat{Y} = 1$ when observing $X = x$ and $A = a$. We denote by $\hat{Y}_f = \hat{Y}_f(x, a)$ the prediction induced by the classifier $f$.*

Group-wise thresholding rules [8, 58] (GWT rules/classifiers or GWTRs over conditional probabilities are of special importance. Consider an appropriate dominating sigma-finite measure $\mu$ on $\mathcal{X}$ (such as the Lebesgue measure for measurable subsets of $\mathbb{R}^d$, $d \geq 1$, or the uniform measure for finite

---

[3]We assume that, whenever needed, the sets considered are endowed with appropriate sigma-algebras, and all functions considered are measurable with respect to the appropriate sigma-algebras.

sets), and suppose that for all $a \in \mathcal{A}$ and $y \in \mathcal{Y}$, the features $X$ have a conditional distribution $P_{X|a,y}$ given $A = a, Y = y$ with a density $dP_{X|a,y}$ with respect to $\mu$. For all[4] $x \in \mathcal{X}$ and $a \in \mathcal{A}$, let $\eta_a(x) = P(Y = 1|X = x, A = a)$.

**Definition 3.2** (GWT classifier). *A classifier $f$ is a GWTR if there are constants $t_a$, $a \in \mathcal{A}$, and functions $\tau_a : \mathcal{X} \to [0,1]$, $a \in \mathcal{A}$, such that for all $x \in \mathcal{X}$ and $a \in \mathcal{A}$*

$$f(x, a) = I(\eta_a(x) > t_a) + \tau_a(x)I(\eta_a(x) = t_a), \tag{1}$$

*where $I(\cdot)$ is the indicator function.*

Clearly, GWTRs choose individuals with the highest conditional probability in each group. This property is a minimal requirement for within-group fairness. For example, a GWT recruitment tool ensures that the most qualified candidates are approved in every protected group. For many independence- and separation-based fairness metrics, the connection between Bayes-optimality and GWTRs has been well-documented in the literature. Corbett-Davies et al. [8] proved that, under demographic parity and predictive equality, the fair Bayes-optimal classifiers are GWTRs with unspecified thresholds. By linking demographic parity and equality of opportunity with cost-sensitive risks, Menon and Williamson [36] further derived the thresholds for fair Bayes optimal classifiers under these two fairness measures. Under perfect demographic parity and equality of opportunity, exact forms of fair Bayes-optimal classifiers were derived in [7] and [43], respectively. More recently, by leveraging the Neyman–Pearson argument from hypothesis testing, Zeng et al. [58] proposed a general framework for deriving fair Bayes-optimal classifiers under independence-and separation-based fairness measures. They have elucidated a direct dependence of the optimal fair thresholds on the level of disparity.

In this paper, we consider predictive parity, which aims to ensure the same positive predictive value among protected groups:

**Definition 3.3** (Predictive Parity). *A classifier $f$ satisfies predictive parity if for all $a \in \mathcal{A}$,*

$$P(Y = 1|\widehat{Y}_f = 1, A = a) = P(Y = 1|\widehat{Y}_f = 1).$$

We follow [5, 58] to use the difference between positive predictive values to measure the degree of unfairness, defining the Difference in Predictive Parities (DPP) of a classifier $f$ as

$$\text{DPP}(f) = \sum_{a \in \mathcal{A}} |P(Y = 1|\widehat{Y}_f = 1, A = a) - P(Y = 1|\widehat{Y}_f = 1)|.$$

## 4 Fair Bayes-optimal Classifiers under Predictive Parity

Since predictive parity is commonly considered under the scenarios where false positives are particularly harmful [30], we study cost-sensitive classification. For a cost parameter $c \in [0,1]^5$, the cost-sensitive 0-1 risk of the classifier $f$ is defined as

$$R_c(f) = c \cdot P(\hat{Y}_f = 1, Y = 0) + (1 - c) \cdot P(\hat{Y}_f = 0, Y = 1).$$

An unconstrained Bayes-optimal classifier for the cost-sensitive risk is any minimizer $f^\star \in \arg\min_f R_c(f)$. A classical result is that all Bayes-optimal classifiers have the form $f^\star(x, a) = I(\eta_a(x) > c) + \tau I(\eta_a(x) = c)$, where $\tau \in [0,1]$ is arbitrary [16, 36].

In the literature, Liu et al. [31] proved that group calibration can be achieved by unconstrained optimization. In fact, the Bayes-optimal score function $\eta_a(x)$ is clearly calibrated with respect to any collection of groups. As a result, when $\eta_a(x)$ is consistently estimated (which can be achieved by unconstrained optimization), the calibration error is well bounded by the excess risk. However, when the training data is biased, we can have

$$P(Y = 1|\eta_a(X) > c, A = a) \neq P(Y = 1|\eta_{a'}(X) > c, A = a') \quad \text{for} \quad a \neq a'$$

In other words, even though the Bayes score $\eta_a$ satisfies perfect multicalibration, it does not satisfy predictive parity. Taking DPP as a constraint, a fair Bayes-optimal classifier is any minimizer of the cost-sensitive risk among fair classifiers:

$$f^\star_{PPV} \in \underset{f : \text{DPP}(f) = 0}{\arg\min} R_c(f). \tag{2}$$

---

[4]To be precise, this conditional density is defined for $\mu$-almost every $x \in \mathcal{X}$; however for simplicity we say for all $x \in \mathcal{X}$. We use this convention without further mentioning through the paper.

[5]When $c = 1/2$, cost-sensitive risk reduces to the usual zero-one risk.

## 4.1 GWT Fair Bayes-Optimal Classifiers under Predictive Parity

We first identify a sufficient condition under which all fair Bayes-optimal classifier under predictive parity are GWTRs.

**Condition 4.1** (Sufficient condition for Bayes-optimal classifiers to be GWTRs).
$$\min_{a \in \mathcal{A}} P(Y = 1 \mid \eta_a(X) \geq c, A = a) \geq \max_{a \in \mathcal{A}} P(Y = 1 \mid A = a).$$

The sufficient condition 4.1 requires that the minimal group-wise positive predictive value $P(Y = 1 \mid \eta_a(X) \geq c, A = a)$ of the unconstrained Bayes-optimal classifier is lower bounded by the maximal proportion of positive labels $P(Y = 1 \mid A = a)$ among groups. In other words, the performances of different groups vary only moderately: with respect to the unconstrained Bayes optimal classifier, the positive predictive value of the worst group—the proportion of $x$ such that $\eta_a(x) \geq c$—should be greater than the overall performance $P(Y = 1 \mid A = a)$ of the best group. Condition 4.1 holds if $P(Y = 1 \mid A = a) \leq c$ for all $a \in \mathcal{A}$, because $P(Y = 1 \mid \eta_a(X) \geq c, A = a) \geq c$.

These conditions are applicable in settings where $c$ is large, such as in credit lending where false positives are more harmful than false negatives, or if $p_{Y\mid a}$, $a \in \mathcal{A}$ are small, such as in job recruitment or school admissions where the number of slots is much smaller than the number of applications. Under this condition, we present our main result.

**Theorem 4.2** (Main result). *Consider the cost-sensitive 0-1 risk with cost parameter c. If Condition 4.1 holds, then all fair Bayes-optimal classifiers under predictive parity are GWTRs. Thus, for all $f_{PPV}^\star$ from (2), there are $(t_a^\star)_{a=1}^{|\mathcal{A}|} \in [0, 1]^{|\mathcal{A}|}$ and functions $\tau_a^\star(x) : \mathcal{X} \to [0, 1]$ such that, for all $x \in \mathcal{X}$ and $a \in \mathcal{A}$,*
$$f_{PPV}^\star(x, a) = I\left(\eta_a(x) > t_a^\star\right) + \tau_a^\star(x) I\left(\eta_a(x) = t_a^\star\right).$$

Unlike for demographic parity or for equality of opportunity, where the fairness constraint is linear with respect to the probability predictions of the classifier $f$ [36], the DPP constraint is non-linear with respect to $f$. As a consequence, previously used theoretical tools such as the Neyman-Pearson argument from hypothesis testing [58] are no longer valid in this case. Instead, we prove the result using a novel constructive argument. When Condition 4.1 is satisfied, for any classifier satisfying predictive parity, which is not a GWTR, we construct a GWTR that satisfies predictive parity and achieves a smaller classification error. As a result, under Condition 4.1, all fair Bayes-optimal classifiers are GWTRs. Overall, the proof of Theorem 4.2 is quite involved, and requires a lot of careful casework and analysis.

## 4.2 Fair Bayes-optimal Classifiers under Predictive Parity do not Need to be Thresholding Rules

Next, we consider the case when the sufficient condition 4.1 does not hold. For simplicity, we consider a binary protected attribute $a \in \{0, 1\}$ with
$$P(Y = 1 \mid \eta_1(x) \geq c, A = 1) < P(Y = 1 \mid A = 0). \tag{3}$$

Our result shows that, under condition (3), there exist class probabilities $p_a$, $a \in \mathcal{A}$, such that no Bayes-optimal classifier under predictive parity is a GWTR.

**Theorem 4.3.** *Suppose that condition (3) holds. Denote $t_1 = \inf\{t : P(Y = 1 \mid \eta_1(X) \geq t, A = 1) > P(Y = 1 \mid A = 0)\}$. Suppose there exist $\delta_1, \delta_2 > 0$ such that $P(c + \delta_1 < \eta_A(X) < t_1 \mid A = 1) = \delta_2 > 0$. Then, for all $p_1 > \frac{2}{2 + \delta_1 \delta_2}$, no fair Bayes-optimal classifier under predictive parity is a GWTR.*

The condition involving the constants $\delta_1, \delta_2 > 0$ ensures that $\eta_1(X)$ has positive probability to be strictly larger than $c$, which is a technical condition needed in the proof. Theorem 4.3 shows that predictive parity may lead to within-group unfairness, whereby the most qualified individuals are predicted to be unqualified, for a better overall accuracy. By definition, predictive parity requires that the qualifications of selected individuals are similar across the protected groups. Suppose there exists a highly qualified minority group in which most individuals are qualified. Selecting the most qualified individuals in this group leads to a very high standard. As a result, many qualified individuals in other majority groups may be predicted to be unqualified using this standard, leading to accuracy loss. Conversely, if we select less qualified individuals in the highly qualified group, the lower standard allows more qualified individuals from the other groups to be selected, and increases accuracy.

---

**Algorithm 1** FairBayes-DPP

---

**Input:** Datasets $S = \cup_{a=1}^{|\mathcal{A}|} S_a$ with $S = \{x_i, a_i, y_i\}_{i=1}^n$ and $S_a = \{x_j^{(a)}, y_j^{(a)}\}_{j=1}^{n_a}$. Cost parameter $c \in [0, 1]$.

**Step 1**: Estimate $\eta_a(x)$ by $\hat{\eta} = f_{\hat{\theta}}$, with $\hat{\theta}$ from (4)

**Step 2**: Find the optimal thresholds..

Define, for all $t$, $\quad \widehat{\mathrm{PPV}}_a(t) = \frac{\sum_{j=1}^{n_a} I(y_j^{(a)} = 1, \hat{\eta}_a(x_j^{(a)}) \geq t)}{\sum_{j=1}^{n_a} I(\hat{\eta}_a(x_j^{(a)}) \geq t)}, \quad \hat{P}(Y = 1 | A = a) = \frac{1}{n_a} \sum_{i=1}^{n_a} y_j^{(a)}$.

**if** $\min_a \widehat{\mathrm{PPV}}_a(c) < \max_a \hat{P}(Y = 1 | A = a)$ **then**
    Warning: Applying FairBayes-DPP may lead to accuracy loss.
**else**
    Let $t_{\min} = \min\{t : \widehat{\mathrm{PPV}}_1(t) \geq \max_a \hat{P}(Y = 1 | A = a)\}$.
    **for** $t \in \mathcal{T} = [t_{\min}, \max_j \hat{\eta}_1(x_j^{(1)})]$ **do**
        **for** $a \in \mathcal{A} \setminus \{1\}$ **do**
            Find $\hat{t}_a(t)$ such that $\widehat{\mathrm{PPV}}_a(\hat{t}_a(t)) \approx \widehat{\mathrm{PPV}}_1(t)$.
        **end for**
        Let $\hat{f}(x, a, t) = \tilde{f}\left(x, a; \hat{t}_1(t), \hat{t}_2(t), ..., \hat{t}_{|\mathcal{A}|}(t)\right) = I\left(\hat{\eta}_a(x) \geq \hat{t}_a(t)\right)$.
        Let $R_c(t) = \frac{1}{n} \sum_{i=1}^n c^{(1-y_i)}(1 - c)^{y_i} I(y_i \neq \hat{f}(x_i, a_i, t))$.
    **end for**
    Find $\hat{t} = \underset{t \in \mathcal{T}_n}{\mathrm{argmin}} R_c(t)$.
    **Output:** $\hat{f}_{PP}(x, a) = I(\hat{\eta}_a(x) \geq \hat{t}_a(\hat{t}))$
**end if**

---

## 5 FairBayes-DPP: Adaptive Thresholding for Fair Bayes-optimality

In this section, we propose the FairBayes-DPP algorithm (Algorithm 1) for fair Bayes-optimal classification under predictive parity. As mentioned, the DPP constraint is non-linear with respect to the classifier $f$, and is also highly non-convex with respect to the model parameters, even if both the classifier $f$ and the risk function are convex with respect to these parameters. In such cases, incorporating fairness constraints as a penalty in the training objective may be hard due to potential local minima. Therefore, we consider a different approach, developing a new two-step plug-in method based on Theorem 4.2. Suppose we observe data points $(x_i, a_i, y_i)_{i=1}^n$ drawn independently and identically from a distribution $\mathcal{D}$ over the domain $\mathcal{X} \times \mathcal{A} \times \mathcal{Y}$.

**Step 1.** In the first step, we apply standard machine learning algorithms to learn the feature- and group-conditional label probabilities $\eta$ based on the whole dataset. Consider a loss function $L(\cdot, \cdot)$ and the function class $\mathcal{F} = \{f_\theta : \theta \in \Theta\}$ parametrized by $\theta$. The estimator of $\eta$ is obtained by minimizing the empirical risk, $\hat{\eta}_a(x) := f_{\hat{\theta}}(x, a)$, where

$$\hat{\theta} \in \underset{\theta \in \Theta}{\mathrm{argmin}} \frac{1}{n} \sum_{i=1}^n L(y_i, f_\theta(x_i, a_i)). \tag{4}$$

Here we use the cross-entropy loss, as minimizing the empirical 0-1 risk is generally not tractable. At the population level, the minimizers of the risks induced by the 0-1 and cross-entropy losses are both the true conditional probability function [37].

**Step 2.** In the second step, we first check the empirical version of Condition 4.1 for the classifier derived in the first step. To be more specific, we divide the data into $|\mathcal{A}|$ parts, according to the value of $A$: for $a \in \mathcal{A}$, $S_a = \{x_j^{(a)}, y_j^{(a)}\}_{j=1}^{n_a}$, where $a_j^{(a)} = a$. Let, for all $t$ for which it is defined,

$$\widehat{\mathrm{PPV}}_a(t) = \frac{\sum_{j=1}^{n_a} I(y_j^{(a)} = 1, \hat{\eta}_a(x_j^{(a)}) \geq t)}{\sum_{j=1}^{n_a} I(\hat{\eta}_a(x_j^{(a)}) \geq t)} \quad \text{and} \quad \hat{P}(Y = 1 | A = a) = \frac{1}{n_a} \sum_{i=1}^{n_a} y_j^{(a)}.$$

We only divide by nonzero quantities here and below. To ensure that the quantities we divide by are nonzero, we restrict to $t_a \in [0, \max_j(\hat{\eta}_a(x_j^{(a)}))]$ when evaluating $\widehat{\mathrm{PPV}}_a(t_a)$. We check whether

$\min_a \widehat{\text{PPV}}_a(c) \geq \max_a \hat{P}(Y = 1|A = a)$.[6] If this is not satisfied, we provide a warning message that applying FairBayes-DPP may lead to accuracy loss, see the discussion after Theorem 4.3. If it is satisfied, we then adjust the thresholds of the classifier aiming for predictive parity. Based on Theorem 4.2, we consider the following deterministic classifiers:

$$\tilde{f}(x, a; t_1, t_2, ..., t_{|\mathcal{A}|}) = I\left(\hat{\eta}_a(x) \geq t_a\right), \tag{5}$$

where $\hat{\eta}$ is the estimate of $\eta$ from the first step, and $t_a$, $a \in \mathcal{A}$, are parameters to learn.

We use the following strategy to estimate $t_a$, $a \in \mathcal{A}$: First, we fix the threshold for the group with $a = 1$, say $t$. The positive predictive value for this group can then be estimated by $\widehat{\text{PPV}}_1(t)$. To achieve predictive parity, we need to find thresholds for the other groups such that the positive group-wise predictive values are the same[7], i.e., find $t_a$, $a = 2, 3, \ldots, |\mathcal{A}|$, such that

$$\widehat{\text{PPV}}_a(t_a) \approx \widehat{\text{PPV}}_1(t), \quad \text{for } a = 2, 3, ..., |\mathcal{A}|. \tag{6}$$

As stated in Lemma A.1, the positive predictive value for each group in the population is always non-decreasing with the thresholds $t_a$ increases. As a consequence, we can search over $t_a$, $a = 2, 3, \ldots, |\mathcal{A}|$, efficiently via, for instance, the bisection method.[8] Correspondingly, we consider the following range of $t$: $\mathcal{T} = [t_{\min}, \max_j \eta_1(x_j^{(1)})]$ with

$$t_{\min} = \min\{t : \widehat{\text{PPV}}_1(t) \geq \max_a \hat{P}(Y = 1|A = a)\}.$$

We denote by $\hat{t}_a(t)$, $a = 2, 3, \ldots, |\mathcal{A}|$, the estimated thresholds given by (6), writing $\hat{t}_1(t) = t$ for convenience. We consider the classifier (5) with these thresholds:

$$\hat{f}(x, a, t) = \tilde{f}\left(x, a; \hat{t}_1(t), \hat{t}_2(t), ..., \hat{t}_{|\mathcal{A}|}(t)\right) = I\left(\hat{\eta}_a(x) \geq \hat{t}_a(t)\right).$$

Lastly, we find $t$ that minimizes the cost-sensitive risk on the training data by searching over a grid $\mathcal{T}_n$ within $\mathcal{T}$:

$$\hat{t} = \underset{t \in \mathcal{T}_n}{\arg\min} \left\{ \frac{1}{n} \sum_{i=1}^n c^{(1-y_i)}(1-c)^{y_i} I(y_i \neq \hat{f}(x_i, a_i, t)) \right\}.$$

Our final estimator of the fair Bayes-optimal classifier is $\hat{f}_{PP} = \hat{f}_{\hat{t}}$. The FairBayes-DPP algorithm is related to the algorithms proposed for other fairness measures in [58], where a binary protected attribute is considered and closed-form optimal thresholds are derived. In contrast, FairBayes-DPP can handle multi-class protected attributes and does not rely on closed-form thresholds. Similar to [58], our algorithm enforces fairness only in the fast second step, where no gradient-based technique is applied. Thus, it is computationally efficient and the non-convexity of fairness constraint is no longer problematic. Our experimental results demonstrate that our method removes disparities and preserves accuracy.

## 6 Experiments

### 6.1 Synthetic Data

We first study a synthetic dataset to compare our method with the true Bayes-optimal fair classifier derived analytically using the true data distribution.

**Statistical model.** Let $X = (X_1, X_2) \in \mathbb{R}^2$ be a generic feature, $A \in \{0, 1\}$ be the protected attribute and $Y \in \{0, 1\}$ be the label. We generate $A$ and $Y$ according to the probabilities $P(A = 1)$, $P(Y = 1|A = 1)$ and $P(Y = 1|A = 0)$, specified below. Conditional on $A = a$ and $Y = y$, $X$ is generated from a bivariate Gaussian distribution $N((2a - 1, 2y - 1)^\top, 2^2 I_2)$, where $I_p$ is the $p$-dimensional identity covariance matrix. In this model, $\eta_a(x)$ has a closed form, and we use it to find the true fair Bayes-optimal classifier numerically under the Condition 4.1. More details about this synthetic model can be found in Section C of Appendix.

---

[6]One could modify this to allow some slack; and perform a formal statistical hypothesis test of our sufficient condition.

[7]Since a sample mean $n^{-1} \sum_{i=1}^n Z_i$ of iid random variables $Z_i$ has a variability of order $O_P(n^{-1/2})$, even if the true predictive parities are equal, the empirical versions may differ by $O_P(n^{-1/2})$. However, in our case we simply find the values $t_a, t$ for which they are as close as possible.

[8]The empirical PPV is only approximately monotonic, but this does not cause problems.

Table 1: Classification accuracy and DPP of the true fair Bayes-optimal classifier and our estimator trained via logistic regression on a synthetic data example. See Section 6.1 for details.

| | THEORETICAL VALUE | | | LOGISTIC REGRESSION | | | |
| | FAIR | UNCONSTRAINED | | FAIRBAYES-DPP | | UNCONSTRAINED | |
| $p$ | ACC | DPP | ACC | DPP | ACC | DPP | ACC |
|---|---|---|---|---|---|---|---|
| 0.2 | 0.814 | 0.000 | 0.814 | 0.049 (0.036) | 0.813 (0.005) | 0.046 (0.037) | 0.813 (0.005) |
| 0.3 | 0.794 | 0.024 | 0.794 | 0.037 (0.029) | 0.794 (0.006) | 0.040 (0.033) | 0.794 (0.005) |
| 0.4 | 0.781 | 0.050 | 0.781 | 0.035 (0.029) | 0.781 (0.006) | 0.054 (0.029) | 0.782 (0.005) |
| 0.5 | 0.775 | 0.078 | 0.777 | 0.042 (0.032) | 0.775 (0.006) | 0.081 (0.036) | 0.777 (0.006) |
| 0.6 | 0.778 | 0.113 | 0.781 | 0.038 (0.031) | 0.778 (0.006) | 0.113 (0.037) | 0.781 (0.006) |

**Experimental setting.** We randomly sample $50,000$ training data points and $5,000$ test data points. In the Gaussian case, the Bayes-optimal classifier is linear in $x$ and thus we employ logistic regression to learn $\eta_1(\cdot)$ and $\eta_0(\cdot)$. We then search over a grid with spacings equal to $0.001$ over the range we identified in Section 5 for the empirically optimal thresholds under fairness. We denote $\widehat{f}$ and $\widehat{f}_{PPV}$ the estimators of the unconstrained and fair Bayes-optimal classifiers, respectively.

We first evaluate the FairBayes-DPP algorithm under the Condition 4.1. We set the cost parameter $c = 0.5$, while $P(A = 1) = 0.3$ and $P(Y = 1|A = 0) = 0.2$. It can be calculated that $P(Y = 1| \eta_0(X) > 0.5, A = 0) \approx 0.613$, using (23) in the Appendix. To consider settings with varied levels of fairness in the population, we vary $p = P(Y = 1|A = 1)$ from 0.2 to 0.6, with the DPP of unconstrained Bayes-optimal classifier grows from 0 to 0.113.

Table 1 presents the classification accuracy and DPP of the true fair Bayes-optimal classifier and FairBayes-DPP trained via logistic regression over 100 simulations[9]. Our first observation is that, under predictive parity, the accuracy of true unconstrained and fair Bayes-optimal classifiers is almost identical, indicating that predictive parity under Condition 4.1 requires a very small loss of accuracy.

Second, our FairBayes-DPP method closely tracks the behavior of the fair Bayes-optimal classifier, controlling the accuracy metric ACC and unfairness metric DPP on the test data effectively. When $|P(Y = 1|A = 1) - P(Y = 1|A = 0)|$ is small, FairBayes-DPP performs similarly to the unconstrained classifier. However, when the data is biased against protected groups and $|P(Y = 1|A = 1) - P(Y = 1|A = 0)|$ is large, FairBayes-DPP mitigates the disparity of the unconstrained classifier effectively, while preserving model accuracy. We further conduct extensive simulations to evaluate the FairBayes-DPP algorithm with different model and training setups, as shown in the Appendix. In particular, we also consider the multi-class protected attribute case.

## 6.2 Empirical Data Analysis

**Dataset.** We test FairBayes-DPP on two benchmark datasets for fair classification: "Adult" [14] and "COMPAS" [27]. For each dataset, we randomly sample (with replacement) 70%, 50% and 30% as the training, validation and test set, respectively. To further test the performance of our algorithm on a large-scale dataset, we conduct experiments on the CelebFaces Attributes (CelebA) Dataset [32].

- *Adult:* The target variable $Y$ is whether the income of an individual is more than \$50,000. Age, marriage status, education level and other related variables are included in $X$, and the protected attribute $A$ refers to gender.

- *COMPAS:* In the COMPAS dataset, the target is to predict recidivism. Here $Y$ indicates whether or not a criminal will reoffend, while $X$ includes prior criminal records, age and an indicator of misdemeanor. The protected attribute $A$ is the race of an individual, "white-vs-non-white".

- *CelebA:* CelebA dataset is a large-scale dataset with more than 200,000 face images, each with 40 attributes (including protected attribute "gender" and other 39 different attributes for prediction tasks). Our goal is to predict the face attributes $Y$ based on the images $X$ and remove bias with respect to gender $A$ from the output.

---

[9]Here, the randomness of the experiment is due to the random generation of the synthetic data.

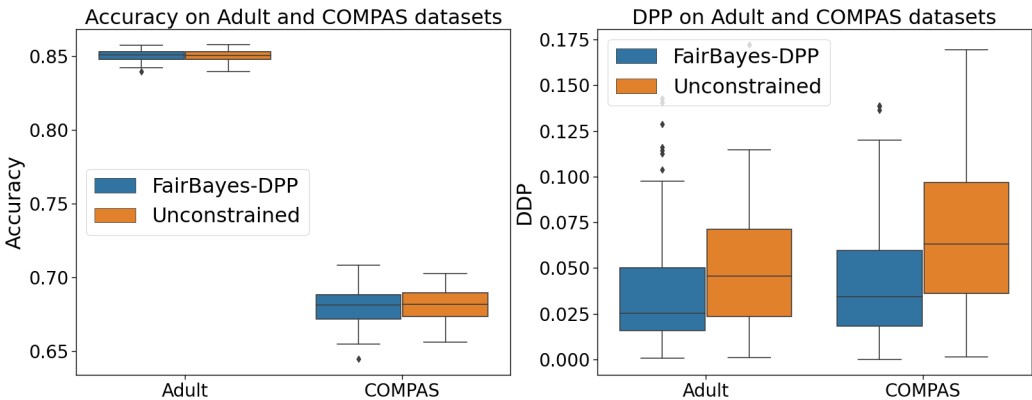

Figure 1: Accuracy and difference in predictive parity on the Adult and COMPAS datasets.

**Experimental setting.** As algorithms for predictive parity are rarely considered in the literature, we use unconstrained learning as a baseline for our experiments. For the "Adult" and "COMPAS" datasets, we adopt the same training setting as in [5, 58]. The conditional probabilities are learned via a three-layer fully connected neural network architecture with 32 hidden neurons per layer. For "CelebA", we apply the training setting from [51]. We learn the conditional probabilities by training a ResNet50 model [23], pretrained on ImageNet [11]. For all the datasets, Over the course of training the model on the training set, we select the one with best performance on the validation set. In addition, we learn the optimal thresholds over the validation set to avoid overfitting. All experiments use PyTorch. We refer readers to the Appendix for more training details, including optimizer, learning rates, batch sizes and training epochs. We repeat the experiment 100 times for the Adult and COMPAS datasets and 10 times for the CelebA dataset.[10]

Figure 1 presents the average performances of FairBayes-DPP and unconstrained learning on the Adult and COMPAS datasets. Our method achieves almost the same accuracy as the unconstrained classifier, and has a smaller disparity. To better compare our fair classifier with the unconstrained one, we use the paired $t$-test to compare the DPP of the proposed algorithm ($\text{DPP}_{Fair}$) and of unconstrained learning ($\text{DPP}_{Base}$). We consider the following one sided test:

$$\mathcal{H}_0 : \text{DPP}_{fair} = \text{DPP}_{Base} \quad \text{v.s.} \quad \mathcal{H}_1 : \text{DPP}_{fair} < \text{DPP}_{Base}.$$

The $p$-values of the tests are $3.90 \times 10^{-4}$ for the Adult dataset and $3.09 \times 10^{-8}$ for the COMPAS dataset. In both cases, these results provide evidence that our FairBayes-DPP achieves a smaller disparity than unconstrained learning.

Finally, we test FairBayes-DPP on the CelebA dataset, Here, we only consider 27 attributes[11] with $0.01 \leq P(Y = 1|M), P(Y = 1|F) \leq 0.99$ in the training, validation, and test sets to ensure that the training, validation and test sample sizes are large enough for each subgroup. We further identify one attribute, "Young", that violates Condition 4.1. We calculate the per-attribute accuracies and DPPs on the test set. Table 2 presents the results of the first six attributes; the remaining results are in the Appendix. As we can see, even for the large-scale CelebA dataset with high dimensional image features, our algorithm mitigates the gender bias effectively, with almost no loss of accuracy.

---

[10]For the Adult and COMPAS datasets, the randomness of the experiment comes from the random selection of the training, validation and test data, as well as the stochasticity of the batch selection in the optimization algorithm. For the CelebA dataset, the randomness is caused by the stochasticity of the optimization method.

[11]Among the 39 attributes, 12 are heavily skewed with $\min(P(Y = 1|M), P(Y = 1|F)) < 0.01$ or $\max(P(Y = 1|M), P(Y = 1|F)) > 0.99$ (where $M$ represents Male and $F$ represents Female) in the training, validation or test set. They are: "5 o'Clock Shadow", "Bald", "Double Chin", "Goatee", "Gray Hair", "Heavy Makeup", "Mustache", "No Beard", "Rosy Cheeks", "Sideburns", "Wearing Lipstick" and "Wearing Necktie".

Table 2: Per-attribute accuracy and DPP of the FairBayes-DPP algorithm and unconstrained optimization.

| ATTRIBUTES | PER-ATTRIBUTE ACCURACY | | PER-ATTRIBUTE DPP | |
| --- | --- | --- | --- | --- |
| | FAIRBAYES-DPP | UNCONSTRAINED | FAIRBAYES-DPP | UNCONSTRAINED |
| ARCHED EYEBROWS | 0.838(0.003) | 0.838(0.003) | 0.027(0.015) | 0.099(0.041) |
| ATTRACTIVE | 0.825(0.002) | 0.826(0.003) | 0.075(0.011) | 0.169(0.016) |
| BAGS UNDER EYES | 0.853(0.002) | 0.852(0.002) | 0.024(0.015) | 0.056(0.034) |
| BANGS | 0.959(0.001) | 0.959(0.001) | 0.007(0.007) | 0.069(0.029) |
| BIG LIPS | 0.706(0.002) | 0.717(0.003) | 0.023(0.015) | 0.115(0.027) |
| BIG NOSE | 0.845(0.002) | 0.847(0.003) | 0.083(0.020) | 0.145(0.023) |

# 7 Summary and Discussion

In this paper, we investigate fair Bayes-optimal classifiers under predictive parity. We prove that when the overall performances of different protected groups vary only moderately, all fair Bayes-optimal classifiers under predictive parity are GWTRs. We further propose a post-processing algorithm to estimate the optimal GWTR. The derived post-processing algorithm removes the disparity in unconstrained classifiers effectively, while preserving a similar test accuracy.

However, when our sufficient condition is not satisfied, the fair Bayes-optimal classifier under predictive parity may lead to within-group unfairness for the minority group. In the current literature, many algorithms directly apply penalized/constrained optimization to impose fairness. Our negative finding, however, is an important reminder that careful analysis is required before employing a fairness measure. The improper use of a measure may result in severe unintended consequences.

# Acknowledgements

Xianli Zeng would like to acknowledge the supported from Shenzhen Research Institute of Big Data (SRIBD); Edgar Dobriban was supported in part by the NSF under award DMS 2046874 (CAREER) and the NSF-Simons Collaboration on the Mathematical and Scientific Foundations of Deep Learning (NSF 2031985); Guang Cheng was supported in part by ONR grant N00014-22-1-2680 and NSF-SCALE MoDL (2134209).

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
