# OpenReview forum: "Fair Bayes-Optimal Classifiers Under Predictive Parity"
_NeurIPS.cc/2022/Conference — NeurIPS 2022 Accept_

### Official Review · Reviewer_aL3t · 2022-07-09

**Rating:** 6
**Confidence:** 4
**Soundness:** 3 good
**Presentation:** 2 fair
**Contribution:** 2 fair

**Summary:**

The authors propose FairBayes-DPP, a post-processing method to achieve the Predictive Parity Fairness metric while maintaining accuracy levels that are comparable to unconstrained classifiers. They also propose a sufficient condition under which all Bayes-optimal classifiers that achieve predictive parity are Group-Wise Thresholding Rules (GWTR), and show that when the condition doesn't hold, it is possible that no Bayes-optimal classifier achieving predictive parity is GWTR, which they argue could lead to within-group unfairness.

**Questions:**

Section 1
1. line 21: "Due to historical bias, vulnerable groups..." Sounds like the authors attribute algorithmic bias necessarily to "problems with the data", and historical biases. However, there are works and ongoing debates about sources of bias that can come from both data, representation, algorithms, etc., so overall there is evidence to suggest it may arise throughout the ML pipeline. You may want to make the phrasing more general.
2. line 34: this may be my ignorance, but why is GWTR a *minimal* requirement for within-group fairness? Could there be any stronger requirement?
3. lines 34-37: these lines already assume some familiarity with post-processing fairness, but no explicit connection or mention of this is included. You may want to explain such connection and provide a little more context.
4. lines 40-41: "sufficiency-based measures such as predictive parity are less commonly considered, possibly due to the complexity of their constraints...". As mentioned above, I see this as a serious omission and lack of consideration of the rest of the literature -- works such as the impossibility result and Pleiss et al. likely deterred researchers from looking to achieve both sufficiency and separation measures in any sense, and separation measures were likely preferred since sufficiency is already favoured by unconstrained optimization. I really do believe a further discussion of such arguments is needed, as well as good replies to why and when one should prefer DPP if at all, and why isn't unconstrained optimization good enough on its own for sufficiency
5. lines 42-45: Following my comment above, I think one of the main arguments against the significance of your work is in Liu et al. (which the authors cite as [30]). Could you provide a more direct answer to the question why isn't unconstrained optimization enough and when would FairBayes-DPP even be useful?

Section 2
6. lines 89-90: "...allowing for perfect predictions that equal the label". This sentence can be made clearer. What do you mean by equal the label?
7. line 91: "...the label information is often unknown for some groups...". Could you provide a citation for this argument? It's a point that was made before. Furthermore, one could presumably use the Y labels used in the training data and held-out-splits. Could you be clearer when making this argument?
8. line 109: "and the later tries to minimize". later should be latter.

Section 3
9. Definition 3.2 is from previous works, and mentioned in Zeng et al. (Bayes-Optimal Classifiers Under Group Fairness) as a classical result. Adding a citation here and discussing its connection to previous works could help.

Section 4
10. Some discussion of cost parameter c and its importance for condition 4.1 could be extended and made earlier (I know you did include some discussion in lines 157-160, but it took me time to realize it is the same c featured above Eq. 2). For example, how should one choose or tune such c?
11. lines 153-155: I'm not sure I understood why you interpret condition 4.1 to mean group performance varies only moderately, or why you discuss average performance, when what is being compared is simply the worst preforming group in terms of positive predictive value against the group with the most positive label presence? I may just be slow, but I think this intuition could be stronger if rephrased.
12. line 183-4: "has positive probability to be strictly larger than c, which is a technical condition needed in the proof". This line could be more convincing if you discussed the implications of this condition and how reasonable it is.

Section 5
13. after step 2: "for all t for which is it defined". Did you discuss anywhere whether you are looking for t values with a grid search in [0,1]? Where are these t values taken from?
14. footnote 4: "perform statistical hypothesis test of our sufficient condition". It may be helpful if you added some guidelines or suggestion on how to perform such a test (e.g., in the appendix), making it easier for others to use directly.


**Limitations:**

The authors did not specifically discuss limitations of their work. However, they did think carefully through a condition under which they may advise to not proceed with the use of their algorithm. I believe a broader context could have been considered in terms of suggesting when and how to use their proposed approach, in particular in relation to other possible fairness criteria. I think the manuscript could be strengthened by further discussion of relation to alternative fairness measures, but do not see it as a major concern.

**Strengths And Weaknesses:**

Strengths:
* The paper is well-written, and does a good job with describing its main findings: connecting DPP with Bayes-optimality, and proposing an efficient post-processing algorithm for achieving DPP without meaningfully trading off accuracy. They further explain their sufficiency condition, and incorporate it directly into their proposed algorithm, FairBayes-DPP.

Weaknesses:
* One might question the novelty and significance of this work in relation to other works studying Bayes-optimality under fairness criteria, post-processing method (in particular FairBayes), and especially given the knowledge we have about calibration error being already bounded by the excess risk for unconstrained classifier (Liu et al. 2019 The Implicit Fairness Criterion of Unconstrained Learning).  I can see the last point being the main challenge to the significance of this work.
* At the moment, the authors suggest that the reason sufficiency measures are understudied in the fairness literature could be the complexity of such constraints or the lack of label information. They seem to completely ignore other considerations such as the impossibility result of Kleinberg et al. (Inherent Trade-Offs in the Fair Determination of Risk Scores), Chouldechova et al. (Fair prediction with disparate impact: A study of bias in recidivism prediction instruments) etc; the more explicit warnings in Pleiss et al. (On fairness
and calibration); or the result showing calibration is favoured by unconstrained optimization. To me, this suggest further consideration of the context and implications of this work could be useful and strengthen this manuscript.
* Some of the implications of the finding could be better discussed: assume the sufficiency condition 4.1 does not hold. Does one necessarily need to look for other fairness criteria? Or is this just another type of fairness-accuracy tradeoff one should consider? What is the relation to Separation measures (Equalized odds, Equality of Opportunity, etc.)? And to Demographic parity? When should one prefer one over the other?
* I believe the authors wrote the manuscript assuming familiarity with post-processing and thresholding works within the fairness literature. I will make suggestions below about how to make it more accessible to those less familiar with this corpus.
* Details such as choice of cost parameter c, thresholds t, etc. could be made clearer.

---

> ### Author Response · Authors · 2022-08-02
> **Author Response to the Reviewer aL3t**
>
> We do appreciate your detailed review and very constructive comments. Due to the page limit, we only answer the most severe concerns, and we have revised our paper according to all your suggestions.
>
> Q2. Why is GWTR a minimal requirement for within-group fairness?
>
> In our paper, an algorithm satisfies within-group fairness if the algorithm prefers the most qualified individuals within each protected group (e.g., Male or Female). As discussed in [1] and [2],  a principal weakness of group fairness measures is that the guarantees are too coarse and may lead to in-group unfairness. For example, consider two protected groups $S_1$ with $A=1$ and $S_0$ with $A=0$. Let us randomly accept every member of $S_1$ and $S_0$ with a fixed probability $p$. While satisfying demographic parity perfectly, the qualified members of both $S_1$ and $S_0$ with large conditional probabilities ($P(Y=1|A=a, X=x)$) will be hurt. With this in mind, we believe that the instance-level comparison in each protected group is necessary for within-group fairness. i.e., we should select the most qualified proportion in each protected group.
>
> Q4. Why and when one should prefer DPP if at all?
>
> We also notice that different fairness constraints may not be compatible with each other. In fact, there are a lot of debates regarding different fairness metrics, and there is no commonly accepted fairness measure. Indeed, the application of fairness-metric is mostly task-dependent. Both separation- and sufficiency-based measures are applied in practice. For example, predictive parity is recommended to be applied in recidivism prediction [3]. In our paper, we do not argue that predictive parity is a better metric for fair classification. However, with its applications in practice, we believe that providing a theoretical benchmark and designing an efficient algorithm for predictive parity is of great practical importance.
>
> Q5.	Why isn't unconstrained optimization enough?
>
> Even though both predictive parity and multi-calibration are sufficiency-based fairness measures, they are different as the formal is conditional on the decision $\hat{Y}_f$ and the latter is conditional on the score function $\hat\eta_a(x)$.
> The unconstrained optimization aims to estimate the Bayes score function $\eta_a(x)$ that always satisfies the sufficiency and multicalibration. However, the induced Bayes decision rule $\eta_a(x)>c$ may not satisfy predictive parity. In general, we have
> $$P(Y=1|\eta_a(X)>c ,A=a)\neq P(Y=1|\eta_a’(X)>c ,A=a’),  a\neq a’.$$
> As a result, in general, unconstrained optimization cannot generate a classifier that satisfies predictive parity. As predictive parity has been applied in recidivism prediction [3], our fairBayes-DPP does have practical values. The simulation study also reveals that FairBayes-DPP can effectively reduce the DPP value over unconstrained optimization.
>
> Q10. Discussion of cost parameter c.
>
> In a cost-protected classification setting, $c$ is a problem-specific parameter (predetermined) rather than a tuning parameter. It is decided by the ratio between loss of false positives and loss of false negatives. For example, in a credit lending setting, the loss of a false negative (decline an applicant who will repay the loan) is the interest rate, and the loss of a false positive (approve an applicant who will default the loan) is the principal of the loan. In this case, the loss of false positives is much more harmful, and the bank will choose a large $c$ to avoid false positives.
>
> Q11. interpretation of condition 4.1.
>
> Thanks for the suggestion. We have rephrased it as follows: the performances of different groups vary only moderately: with respect to the unconstrained Bayes optimal classifier, the positive predictive value of the worst group---the proportion of $x$ such that $\eta_a(x)\geq c$---should be greater than the overall performance $P({Y=1|A=a})$ of the best groups.
>
> Q12.  Discussion about technique condition:
>
> In fact, this is a very weak condition requiring that at least some of the individuals in the majority group are qualified. When $P(\eta_1(X)>c)=0$, we have all individuals in the majority groups are not qualified, which rarely happens in practice. In this very special case, note that the unconstrained Bayes optimal classifier is $I(\eta_a(x)>c)$. Hence, the optimal strategy may be just to decline all the applications, and this is still a GWTR that satisfies predictive parity.
>
> [1] Dwork, C., Hardt, M., Pitassi, T., Reingold, O., \& Zemel, R. (2012). Fairness through awareness. In Proceedings of the 3rd innovations in theoretical computer science conference.
>
> [2] H\'{e}bert-Johnson, U., Kim, M., Reingold, O., \& Rothblum, G. (2018). Multicalibration: Calibration for the (computationally-identifiable) masses. In International Conference on Machine Learning, PMLR.
>
> [3] Dieterich, W., Mendoza, C., \& Brennan, T. (2016). COMPAS risk scales: Demonstrating accuracy equity and predictive parity. Northpointe Inc.

---

> > ### Comment · Reviewer_aL3t · 2022-08-08
> > **Thank you for your response!**
> >
> > Thank you for providing clarifications to my questions.
> >
> > I found all of the answers helpful, and would like to emphasize that I understand Q4 may be somewhat out of scope for this manuscript, but I still think that if the authors can shed any light on this discussion, that would be of help to the community.
> >
> > At this stage it seems to me like the main objections to acceptance of this paper are
> > * Reviewer AKkJ's concern 1 that I understand, but does not subtract from the value I believe the work already has, and
> > * Reviewer rrxc's concerns with the motivation of the paper. I do understand the value and importance in the motivation for the work.
> >
> > **I would therefore recommend its acceptance**.
> >
> > I hope the authors will use their proposed answers to mine and other reviewers' questions in the final version of the manuscript (at least in the appendix), as I believe they could help future readers.

---

> > > ### Author Response · Authors · 2022-08-09
> > > **Thanks for the very encouraging comments**
> > >
> > > Dear Referee,
> > >
> > > We do appreciate your positive and very encouraging comments. We will add these helpful discussions in the final version to better validate our paper.
> > >
> > > Best,
> > >
> > > Authors

---

### Official Review · Reviewer_AKkJ · 2022-07-11

**Rating:** 6
**Confidence:** 3
**Soundness:** 3 good
**Presentation:** 3 good
**Contribution:** 3 good

**Summary:**

The authors investigate the Bayes-optimal classifier for the fair classification with the predictive parity as a fairness constraint. They analyze whether the Bayes-optimal classifier forms a group-wise thresholding rule. As a result, they reveal the sufficient condition under which the Bayes-optimal classifier forms a GWTR. They also develop the algorithm for constructing a GWTR under the predictive parity constraint. The experimental results demonstrate that the proposed algorithm achieves fairer classifications than the unconstrained one.

**Questions:**

See strengths and weaknesses.

**Ethics Review Area:**

["I don’t know"]

**Limitations:**

From the viewpoint of potentially negative impact, it is better to include the suggestion of the other fairness definition, which should be chosen when Condition 4.1 is not satisfied.

**Strengths And Weaknesses:**


(Strengths)
- Clearly written and technically sound.
- The condition of forming the Bayes-optimal classifier as a GWTR is interesting and crucial for understanding the predictive parity constraint.
- The authors give a reasonable algorithm for learning a GWTR while ensuring predictive parity.

(Weakness)
- Similar works investigating the fair Bayes-optimal predictor are motivated to clarify the cost of enforcing the fairness definitions, including the demographic parity and equalized odds. This work is somewhat weak because it cannot reveal the cost of the predictive parity.
- The result is just a sufficient condition is not a necessary and sufficient condition. The complete characterization of this phenomenon is still unfulfilled.
- Empirical comparison is somewhat weak. It lacks comparison with the existing sufficiency-based fair learning techniques.


This paper is clearly written and technically sound. The condition of forming the Bayes-optimal classifier as a GWTR is an interesting direction for understanding the predictive parity constraint. My current recommendation is acceptance.


The motivation of the existing works investigating the fair Bayes-optimal classifiers mostly comes from clarifying the cost of enforcing a fairness constraint. To this end, these works reveal the difference in accuracy between the unconstrained and constrained versions of the Bayes-optimal classifier. While clarifying the cost is well-motivated, the analyses in this paper are insufficient to reveal the cost of enforcing the predictive parity. It is better to include a discussion about this point.


The proposed method, FairBayes-DPP, is constructed reasonably. However, I am concerned about its statistical efficiency. Some researchers reveal the statistical efficiency of fair learning with the sufficient-based fairness definitions, including
- C. Jung et al. Moment Multicalibration for Uncertainty Estimation. COLT2021.
- E. Shabat et al. Sample Complexity of Uniform Convergence for Multicalibration.

I wonder whether the proposed method significantly improved from the methods analyzed in these papers. The problem setting of these papers might be somewhat different from this paper. Nevertheless, it is better at least to discuss the differences.

In the experiment, the authors only compare their method with the unconstrained one. I thus have concern that the proposed method underperforms the existing ones, including those introduced in this paper as "sufficient-based" ones, and additionally one in the following paper:
- U. Hebert-Johnson et al. Multicalibration: Calibration for the (Computationally-Identifiable) Masses. ICML2018.


Minor issues:
- Line 537 in the appendix: I cannot follow the inequality. Why the function $\tau \to (w_{a1}(t)+\tau v_{a1}(t))/(w_a(t)+ \tau v_a(t))$ is non-decreasing?
- Line 542 in the appendix: The definitions of $T_0$ and $T_1$ looks the same. Maybe, it is typo.

---

> ### Author Response · Authors · 2022-08-02
> **Author Response to the Reviewer AKkJ**
>
> We appreciate your careful review and encouraging comments.  We will reply to your concerns in the following:
>
> Concern 1: The cost of predictive parity.
>
> When condition 4.1 holds, based on our theorem 4.2, we need to adjust thresholds for each group to achieve predictive parity.  The change of thresholds is always at the cost of accuracy loss.  In fact, as shown in table 1 as well as Figures 2 to 5 in the appendix, there is a non-vanishing gap between the accuracy of the unconstrained classifier and the classifier that satisfies predictive parity.  Moreover, under the condition of Theorem 4.3, we have shown that the fair Bayes-optimal classifier is not a GWTR.
> This indicates that imposing predictive parity either leads to within-group unfairness, or results in a sacrifice of accuracy.
>
> Concern 2: Compare FairBayes-DPP with multicalibration.
>
> Thanks for the valuable suggestion and for bringing our attention to the multi-calibration literature.  We want to clarify that, even though both predictive parity and multi-calibration are sufficiency-based fairness measures, they are different in that the former is conditional on the decision $\widehat{Y}_f$, and the latter is conditional on the score function $\widehat\eta_a(x)$.
>
> Multicalibration aims to derive a calibrated estimator of $\eta_a(x)$, over different groups.
> However, having a calibrated estimator over even an arbitrarily complex set of groups does not seem enough.
> In fact, even the Bayes-optimal score function $\eta_a(x)$ (which is obviously calibrated with respect to any collection of groups),
> may not lead to a classifier
> satisfying predictive parity.
> In general, when the training data is biased, we can have
> $$P(Y=1|\eta_a(X)>c ,A=a)\neq P(Y=1|\eta_{a’}(X)>c ,A=a’), \ \ \ \  a\neq a’.$$
> In other words, even though the derived estimator satisfies perfect multicalibration, it does not satisfy predictive parity.
> Our FairBayes-DPP is targeting these inequalities inherited from the training data.  We generate a fair classifier that satisfies predictive parity by adjusting thresholds for different protected groups.
>
> To summarize, as multicalibration aims to estimate $\eta_a(x)$ in a multicalibrated way, rather than generating a classifier that satisfies predictive parity, we do not think it is reasonable to compare our FairBayes-DPP with multicalibration methods.  Instead, we believe that, in future work, multicalibration can be incorporated with our FairBayes-DPP: applying multicalibration to generate an unbiased estimator for $\eta_a$ and then applying our second step to derive group-wise thresholds over estimated conditional probabilities.
>
>
> Concern 3.  The computational complexity of FairBayes-DPP:
>
> We also noticed that researchers had revealed the computation complexity of multicalibration methods.  For multicalibration, we require that $|\widehat\eta_a-{\eta}_a|$ is close to $0$ for any given level of $\eta_a$. However, predictive parity only necessitates the set $\\{x: \widehat{\eta}_a(x)>\widehat{t}_a\\}$ is close to the set $\\{x:\eta_a(x)>t_a\\}$. In other words, it only focuses on the sign (compared to the thresholds) rather than the value of the prediction.  Moreover, as the PPV values are non-decreasing with respect to the thresholds, for a given PPV value, the corresponding thresholds can be learned very efficiently.  For the Adult dataset with the same computation environment, the average running times are 50.8s for the unconstrained classifier and 52.6s for our FairBayes-DPP, respectively, indicating the efficiency of our algorithm.
>
>
> Minor concerns:
>
> (1) For fixed $t$, let $h(\tau)=\frac{w_{a1}(t)+\tau v_{a1}(t)}{w_a(t)+\tau v_a(t)}$. We have $h(\tau)$ is a non-increasing (not non-decreasing) function of $\tau$ since,
> \begin{eqnarray*}\frac{d h(\tau)} {d\tau}&=&  \frac{v_{a1}(t)(w_a(t)+\tau v_a(t))-v_a(t)(w_{a1}(t)+\tau v_{a1}(t))}{(w_a(t)+\tau v_a(t))^2}\\
> &=& \frac{v_{a1}(t)w_a(t)-v_a(t)w_{a1}(t)}{(w_a(t)+\tau v_a(t))^2}\\
> &\leq&  \frac{tv_{a}(t)w_a(t)-tv_a(t)w_{a}(t)}{p_{Y|a}(w_a(t)+\tau v_a(t))^2}=0.\\
> \end{eqnarray*}
> Here, the third inequality holds since $v_a(t)\geq 0$, $
> \frac{p_{Y|a}w_{a1}(t)}{w_{a}(t)}>t$ and
> $
> \frac{p_{Y|a}v_{a1}(t)}{v_{a}(t)}=t.
> $
>
>
> (2) Yes, we deleted one of the definitions in the revision.
>
>
> [1] H\'{e}bert-Johnson, U., Kim, M., Reingold, O., \& Rothblum, G. (2018, July). Multicalibration: Calibration for the (computationally-identifiable) masses.  In International Conference on Machine Learning (pp. 1939-1948).  PMLR.

---

> > ### Comment · Reviewer_AKkJ · 2022-08-08
> > **Thanks for your reply**
> >
> >
> > I thank the authors for the response.
> >
> > > Concern 1
> >
> > I'm afraid that the authors do not address my concern. I pointed out that the authors do not reveal the closed-form of the Bayes optimal classifier as, for example, A. K. Menon and R. C. Williamson did. If the closed-form is obtained, we can discuss the cost of fairness by comparing it with the Bayes optimal classifier of the unconstrained problem. Also, through the closed-form and further analyses, they construct an algorithm for estimating the trade-off frontier between accuracy and fairness. I commented that the current analyses are insufficient for conducting such a discussion.
> >
> > > Concern 2
> >
> > I understand the difference from the multi-calibration. I thank the authors for the detailed comparison. I think it is better for the reader to involve this discussion in the main body.
> >
> > > Concern 3
> >
> > I've not concerned about the computational complexity but statistical efficiency, meaning how the estimation error behaves along with the sample size. Could the authors comment on this point?
> >
> > Even after reading the response, I'd like to keep my score accepted.

---

> > > ### Author Response · Authors · 2022-08-09
> > > **Response to Further Concerns (Part I)**
> > >
> > > Thanks a lot for your further comments. We are happy to clarify your further concerns.
> > >
> > > $\\textbf{Concern 1}$: I'm afraid that the authors do not address my concern. I pointed out that the authors do not reveal the closed-form of the Bayes optimal classifier as, for example, A. K. Menon and R. C. Williamson did. If the closed-form is obtained, we can discuss the cost of fairness by comparing it with the Bayes optimal classifier of the unconstrained problem. Also, through the closed-form and further analyses, they construct an algorithm for estimating the trade-off frontier between accuracy and fairness. I commented that the current analyses are insufficient for conducting such a discussion.
> > >
> > > $\\textbf{Response}$: In general, the closed-form of the Bayes optimal classifier may not exist due to the non-linearity of the constraint.
> > > We will show that this is the case in perhaps the simplest and most natural example, that of a two-class Gaussian classification problem with a binary protected attribute. Note that in this case the Bayes-optimal classifiers for other constraints such as demographic parity are easy to find in a closed form. However, this is not the case for predictive parity. Thus, the type of results from MW18 cannot be obtained for predictive parity.
> > >
> > >
> > > Indeed, let us consider the synthetic model discussed in the appendix: with $A\in\\{0,1\\}$, $Y\in\\{0,1\\}$, the distribution of $(X,A,Y)$ is as follows,
> > >
> > > (1) For $a\in\\{0,1\\}$, $P(A=a)=p_a$ and $P(Y=1|A=a)=1-P(Y=0|A=a)=p_{Y|a}$;
> > >
> > > (2) For $(a,y)\in\\{0,1\\}^2$, $X|A=a,Y=y\sim \mathcal{N}(\mu_{a,y},\sigma^2I_2)$ with $\mu_{a,y}=(2a-1,2y-1)^\top$.
> > >
> > > Let $c=1/2$ and consider the GWTR $f_{t_1,t_0}$ such that for $a\in\\{0,1\\}$ and all $x\in\mathcal{X}$,
> > > $f_{t_1,t_0}(x,a)=I(\eta_a(x)>t_a)$. As we show in the appendix,
> > >  $f_{t_1,t_0}$ satisfies predictive parity if
> > > $$\frac{p_{Y|1}\bar\Phi\left(\frac{\sigma\log(q_1(t_1))}{2}-\frac{1}{\sigma}\right)}{(1-p_{Y|1})\bar\Phi\left(\frac{\sigma\log(q_1(t_1))}{2}+\frac{1}{\sigma}\right)}=\frac{p_{Y|0}\bar\Phi\left(\frac{\sigma\log(q_0(t_0))}{2}-\frac{1}{\sigma}\right)}{(1-p_{Y|0})\bar\Phi\left(\frac{\sigma\log(q_0(t_0))}{2}+\frac{1}{\sigma}\right)}.
> > > $$
> > > where $q_a(t)=\frac{t(1-p_{Y|a})}{(1-t)p_{Y|a}}$ and $\bar\Phi(t)=1-\Phi(t)$ with $\Phi(t)$ the cumulative distribution function of the standard normal distribution. Moreover, the misclassification rate of $f_{t_1,t_0}$ is
> > > $$R(f_{t_1,t_0})=\sum_{a\in\\{0,1\\}} p_a[(1-p_{Y|a})P(\eta_a(X)\ge  t_a|A=a,Y=0)+ p_{Y|a}P(\eta_a(X)<t_a|A=a,Y=1)]=\sum_{a\in\\{0,1\\}}p_a\left[p_{Y|a}\Phi\left(\frac{\sigma\log(q_a(t_a))}{2}-\frac{1}{\sigma}\right)+(1-p_{Y|a})\bar\Phi\left(\frac{\sigma\log(q_a(t_a))}{2}+\frac{1}{\sigma}\right)\right].
> > > $$
> > > Consider the Lagrange multiplier $\\lambda$, and the penalized loss
> > > $$L(t_1,t_0,\\lambda)=R(f_{t_1,t_0})+\\lambda \left(\frac{p_{Y|1}\\bar\\Phi\left(\frac{\sigma\log(q_1(t_1))}{2}-\frac{1}{\sigma}\right)}{(1-p_{Y|1})\bar\Phi\left(\frac{\sigma\log(q_1(t_1))}{2}+\frac{1}{\sigma}\right)}-\frac{p_{Y|0}\bar\Phi\left(\frac{\sigma\log(q_0(t_0))}{2}-\frac{1}{\sigma}\right)}{(1-p_{Y|0})\bar\Phi\left(\frac{\sigma\log(q_0(t_0))}{2}+\frac{1}{\sigma}\right)}\right).$$
> > > Denoting $s_{11}=\frac{\sigma\log(q_1(t_1))}{2}-\frac{1}{\sigma}$, $s_{10}=\frac{\sigma\log(q_1(t_1))}{2}+\frac{1}{\sigma}$, $s_{01}=\frac{\sigma\log(q_0(t_0))}{2}-\frac{1}{\sigma}$ and $s_{00}=\frac{\sigma\log(q_0(t_0))}{2}+\frac{1}{\sigma}$, the first order condition is the following equation system:
> > > $$
> > > p_1(1-p_{Y|1})f(s_{10}) +p_1p_{Y|1}f(s_{11})+\lambda\frac{p_{Y|1}[f(s_{11})\bar\Phi(s_{10})-\bar\Phi(s_{11})f(s_{10})]}{(1-p_{Y,1})\bar\Phi^2(s_{0,1})}=0;$$
> > > $$p_0(1-p_{Y|0})f(s_{00}) +p_0p_{Y|0}f(s_{01})+\lambda\frac{p_{Y|0}[f(s_{01})\bar\Phi(s_{00})-\bar\Phi(s_{01})f(s_{00})]}{(1-p_{Y,0})\bar\Phi^2(s_{0,0})}=0;$$
> > > $$\frac{p_{Y|1}\bar\Phi(s_{11})}{(1-p_{Y|1})\bar\Phi(s_{10})}-\frac{p_{Y|0}\bar\Phi(s_{01})}{(1-p_{Y,0})\bar\Phi(s_{00})}=0.$$
> > > Note that $\Phi(t)$ does not have an explicit form and the equation system can only be solved numerically with no closed-form solution.
> > >
> > >
> > > However, we think that  we can address some of the problems raised by the reviewer numerically, instead of theoretically.
> > > We can discuss the cost of fairness by comparing the numerical results obtained by our method with the Bayes optimal classifier of the unconstrained problem.
> > > NOTE: for this, our theoretical results are still extremely useful, as they reduce the problem of searching over all classifiers (a big class) to just searching over a finite number of thresholds that satisfy a certain constraint.
> > >
> > > Also,
> > > we can construct an algorithm for estimating the trade-off frontier between accuracy and fairness numerically.
> > > Thus, we believe that the current analyses are  sufficient for conducting such a discussion numerically.
> > > Moreover, it is in general impossible to obtain exact theoretical results.
> > > Thus, we believe that we have solved the problem as well as it is possible.

---

> > > ### Author Response · Authors · 2022-08-09
> > > **Response to Further Concerns (Part II)**
> > >
> > >
> > > $\\textbf{Concern 2}:$ I understand the difference from the multi-calibration. I thank the authors for the detailed comparison. I think it is better for the reader to involve this discussion in the main body.
> > >
> > > $\\textbf{Response}:$ Thanks for the suggestion, we will discuss it in the final version of the paper.
> > >
> > > $\\textbf{Concern 3}:$ I've not concerned about the computational complexity but statistical efficiency, meaning how the estimation error behaves along with the sample size. Could the authors comment on this point?
> > >
> > > $\\textbf{Response}:$ This is indeed an extremely important question.
> > > However, we feel that its full investigation requires a separate follow-up paper, as it is a very challenging problem.
> > > In our algorithm, there are two sources of estimation errors (1) estimation error for estimating $\\eta_a$ and (2) estimation error in deciding $t_a$. The former relies on the sample size of the smallest protected group and the regularity of $\eta_a$, which is well documented in the statistical literature ([1],[2]). For the latter one, the procedure of estimating $t_a$ can be viewed as a generalization of level set estimation with a given level ([3],[4]). This estimation error depends on the size of the smallest group and the behavior of $\eta_a$ near the boundary $t_a^\\star$ (e.g. the $\\gamma$ -exponent condition ([4])).
> > > This gives a way to bound the sample complexity; however it is a separate follow-up project to do this work. It will take at least one, if not more papers, and perhaps even a year or so, to fully solve this problem.
> > >
> > >
> > >  [1] Audibert, J. Y., \& Tsybakov, A. B. (2007). Fast learning rates for plug-in classifiers. The Annals of statistics, 35(2), 608-633.
> > >
> > >  [2]0 Kim, Y., Ohn, I., \& Kim, D. (2021). Fast convergence rates of deep neural networks for classification. Neural Networks, 138, 179-197.
> > >
> > >  [3] Rigollet, P., \& Vert, R. (2009). Optimal rates for plug-in estimators of density level sets. Bernoulli, 15(4), 1154-1178.
> > >
> > >  [4] Lei, J., Robins, J., \& Wasserman, L. (2013). Distribution-free prediction sets. Journal of the American Statistical Association, 108(501), 278-287.

---

### Official Review · Reviewer_rrxc · 2022-07-14

**Rating:** 4
**Confidence:** 4
**Soundness:** 3 good
**Presentation:** 3 good
**Contribution:** 2 fair

**Summary:**

This paper studies predictive parity, which is a kind of sufficiency-based fairness and requires the same probability of positive prediction. This paper makes two major contributions:1) Bayes-optimal classifiers satisfying predictive parity may or may not be 66 group-wise thresholding rules (GWTRs) 2) propose the FairBayes-DPP algorithm for binary fair classification, which is a post-processing  algorithm and to estimate the fair Bayes-optimal classifier under predictive parity This paper also provides the experiments to support the proposed method.

**Questions:**

1. This paper should provide more reasons why the proposed method is suitable for this problem.
2. I would suggest adding more introduction and discussion about the GWTR.
3. This paper should explore baselines to demonstrate the effectiveness of the proposed method.


**Ethics Review Area:**

["I don’t know"]

**Strengths And Weaknesses:**

**Strengths**
1.The fairness studied in the paper is interesting. The predictive parity is of practical significance.
2. The theoretical analysis in this paper seems sound to me. The finding that “ the optimal fair classifiers may or may not be a GWTR, depending on the data distribution” is interesting and insightful. And it would be a good addition to the fairness community.
3. This paper is easy to follow.

**Weaknesses**
1. The motivation of this paper is not clear.
    - I do not agree with the claim in this paper that “such as predictive parity are much less studied”(Line 5). As far as I know and also pointed out in this paper, there are tons of paper studies such fairness.
    - The claim that “sufficiency-based measures such as predictive parity are less commonly considered, possibly due to the complexity of their constraints”(Line 40-41) is wrong to me. From Definition 3.3, we can see that the predictive parity is quite similar to separation-based measures(demographic parity), thus I don’t agree such fairness constraints is complicated.
    - The proposed methods FairBayes-DPP is not well motivated. This paper should provide more reasons why the proposed method is suitable for this problem.
2. The preliminary about GWTR is not sufficient. The claim that “for many fairness metrics, the optimal fair classifiers are GWTRs” is not common knowledge to me. Thus I would suggest adding more introduction and discussion about the GWTR.
3. The experiments are not convincing to me. For baselines, one simple way is using the equation in Definition 3.3 as a regularizer. However, this paper does not compare any baselines, which can not show the effectiveness of the proposed method.

Overall, this paper is not well motivated and the experiments are not convincing. This paper in its current form is not ready to publish. Thus I recommend rejection at this time. And I believe this paper is a good addition to the fairness community if the author fixed the concerns.

---

> ### Author Response · Authors · 2022-08-02
> **Author Response to the Reviewer rrxc**
>
> We do appreciate your careful review and valuable comments/suggestions. In the following, we will first summarize your concerns and then provide our clarification.
>
> Concern 1. Authors may have wrongly stated that there is little literature discussing predictive parity, and this may lead to an unclear motivation of the paper.
>
> We agree that predictive parity is widely applied to measure the fairness of a given algorithm. However, to our knowledge, there is no existing theoretical result concerned with Bayes-optimality for predictive parity. Moreover, only very limited literature designs fair algorithms to satisfy predictive parity. Note that to measure whether an algorithm satisfies predictive parity and to design an algorithm to satisfy predictive parity are two different goals. As a result, our derivation of the theoretical benchmark and design of an efficient algorithm for predictive parity has practical value. We will make this claim more precise in the revision.
>
> Concern 2. Difference between sufficiency-based measures and independence/separation-based measures.
>
> We agree that, on a first look, the form of predictive parity may appear to be similar to demographic parity and equality of opportunity.
> However, they are essentially different. As mentioned in [1], the constraints for independence and sufficiency-based measures are linear in the model's predicted probabilities. Moreover, when the model output is convex with respect to model parameters, the constraints are also convex. However, for predictive parity, as the prediction $\hat{Y}$ is conditioned upon, the constraint is non-linear in the predicted probabilities and may be highly non-convex with respect to model parameters. For illustration, let us consider the binary classification problem with binary protected attribute $A\in\{0,1\}$. We compare predictive parity with demographic parity. To be specific, a classifier $f$ satisfies demographic parity if
> $$P(\hat{Y}\_f=1|A=1)=P(\hat{Y}\_f=1|A=0),$$
> and it satisfies predictive parity if
> $$P(Y=1|\hat{Y}\_f=1,A=1)= P(Y=1|\hat{Y}\_f=1,A=0).$$
> With the same notation in the paper, the difference in demographic parity (DDP) can be expressed as
> $$
> DDP = P(\hat{Y}\_f=1|A=1)-P(\hat{Y}\_f=1|A=0)
> =\int f(x,1)P_{X|A=1}(dx)-\int f(x,0)P_{X|A=0}(dx).$$
> which is linear in $f$. However, for predictive parity, the difference in positive predictive value (DPP) can be expressed as,
> $$DPP = P(Y=1|\hat{Y}_f=1,A=1)- P(Y=1|\hat{Y}_f=1,A=0)
> =\frac{P(\hat{Y}_f=1,A=1,Y=1)}{P(\hat{Y}_f=1,A=1)}-\frac{P(\hat{Y}_f=1,A=0,Y=1)}{P(\hat{Y}_f=1,A=0)}.
> $$
> As we can see, $\hat{Y}_f$ appears in both nominator and denominator; thus, the constraint is non-linear with respect to $f$. This non-linearity may lead to non-convex constrained optimization, and also to a much more challenging algorithm design and analysis.
>
> Concern 3. The motivation of FairBayes DPP.
>
>  Our FairBayes-DPP is motivated by our theoretical result. In theorem 4.2, we have provided a sufficient condition under which the fair Bayes optimal classifier is a GWTR over the conditional probabilities. With this result, it is a natural idea to consider a plug-in method that generates a GWTR over the estimated conditional probabilities. Our second step for searching the thresholds is well motivated by the property of the fair Bayes optimal classifier: it achieves the highest accuracy among all classifiers given any constraint of predictive parity. Moreover, we have emphasized the advantages of FairBayes-DPP in the paper. First, as the positive predictive value is non-decreasing with respect to the threshold, for a given PPV value, the desired thresholds can be efficiently searched over using the Bisection method. Second, as the fairness constraint is only handled in the second step, where we used Bisection and grid search, no gradient-based optimization method is applied. As a result, we can effectively handle the non-convexity of the fairness constraint.
>
> Concern 4. The introduction and preliminary about GWTR.
>
> We appreciate your valuable suggestions. In the revision, we have provided a more comprehensive literature review after definition 3.2 to discuss Bayes-optimality and GWTR.
>
> Concern 5. Comparison with other baseline methods.
>
> As we mentioned above, although predictive parity is widely used to measure whether an algorithm is fair, there is little existing literature to design an algorithm aiming for predictive parity. Moreover, as indicated in [2], the sufficiency-based measure is implicitly favored by unconstrained optimization. We thus compare our method with unconstrained optimization in the paper.
>
> [1] Menon, A. K., \& Williamson, R. C. (2018). The cost of fairness in binary classification. In  Conference on Fairness, Accountability and Transparency (pp. 107-118). PMLR.
>
> [2] Liu, L. T., Simchowitz, M., & Hardt, M. (2019). The implicit fairness criterion of unconstrained learning. In International Conference on Machine Learning (pp. 4051-4060). PMLR.

---

### Official Review · Reviewer_m8AG · 2022-07-20

**Rating:** 5
**Confidence:** 3
**Soundness:** 3 good
**Presentation:** 3 good
**Contribution:** 2 fair

**Summary:**

This paper considers the Predictive Parity (PP) notion of fairness and conducts theoretical and empirical analysis. In particular, the paper considers the form of fair Bayes-optimal classifier as group-wise threshold-based classifiers (GWTRs), and provides sufficient condition for GWTRs to be optimal PP classifier. The analysis w.r.t. situations where such condition is or is not violated are also provided.

**Questions:**

### Q1: can authors share some insights regarding the potential necessary condition (or, the difficulty behind such analysis)?

(as detailed in Section `Strengths and Weakness`)

### Q2: w.r.t. the term "within-group fairness"

While I understand the phenomenon described in Section 4, I am a little bit confused about the role played by "within-group fairness". In PP notion of fairness, the definition is w.r.t. the proportion of real positive labels among predicted positives/negatives, there is no detailed characterization of instance-level qualification comparison within each group. I understand that the paper is not indicating the instance-level comparison, but personally I think it would be clearer if the authors can avoid such kind of potential misunderstandings.

### Q3: Equation 3

Is there a typo on $\eta_a(x)$ (i.e., it should be $\eta_1(x)$ since A = 1 is conditioned), or, there is a typo on the value assignment of $A$ (i.e., for left-hand side, A = a, but for right-hand side A = a')? From the context, I am not sure how to parse Equation 3. A clarification would be very helpful.

### Q4: w.r.t. recommended action in Algorithm 1

In lines 211-213, it is mentioned that if Condition 4.1 is not satisfied, the authors recommend considering other fairness notions. This is a little hard to parse. If Condition 4.1 is a sufficient condition, then the violation of it should not be advise us to go for a different fairness notion. After all, Condition 4.1 is not a necessary condition, the violation of which directly indicate the impossibility of pursuing PP.

**Limitations:**

It would be desirable if the paper can provide some discussion w.r.t. necessary condition beyond the sufficient condition 4.1 to give a whole picture of the problem at hand.

Minor typo: line 310 [man] algorithms

**Strengths And Weaknesses:**

Overall, the paper is relatively well-written and easy to follow. The analysis provides connection between the cost-sensitive GWTRs and fair Bayes-optimal for binary classification. A two-stage algorithm is also proposed and evaluated empirically.

The weakness of the paper comes from a potential worry about the takeaway msg. It would be very helpful if the authors can also share the insight (if there are some) behind the possibility of deriving a necessary condition for GWTRs to be the fair Bayes optima. The paper provides sufficient condition, which is relatively intuitive. Condition 4.1 essentially requires that the output score $\eta_a(x)$ assigned to $(a, x)$ should not be too low (across the group) such that the classifier can easily achieve PP (e.g., the trivial case as an extreme). Here, it is natural to wonder what, if there is one, would be the necessary condition? Although this is not a trivial question, it would be nice to provide some analysis on it, especially considering the fact that there is an algorithm proposed to handle cases where Condition 4.1 is not satisfied.

---

> ### Author Response · Authors · 2022-08-02
> **Author Response to the Reviewer m8AG**
>
> Thanks for your constructive comments and suggestions. We provide answers to your questions below.
>
> Q1: Potential necessary condition.
>
> We have implicitly provided a necessary condition in Section 4.2. More specifically, we have proved a negative result (Theorem 4.3) that, under (another) sufficient condition, the fair Bayes optimal classifier concerning predictive parity is not a GWTR. This negative result indeed  leads to a necessary condition (the opposite condition of Theorem 4.3) for the fair Bayes optimal classifier under predictive parity being a GWTR.
> The conditions for sufficiency and necessity (for binary protected attribute) are the complements of each other, under one more condition regarding the group size ($p_a$) specified in Theorem 4.3.
>
> Q2: w.r.t. the term "within-group fairness"
>
>  We agree that, in the definition of predictive parity, there is no instance-level qualification comparison within each group. However, as discussed in [1] and [2],  only requiring group-level parity is insufficient for a fair algorithm. For example, if we consider a random selection among all applicants, then this strategy satisfies demographic parity. However, this random strategy will hurt the qualified members in each protected group (individuals with large conditional probabilities ($P(Y=1|A=a, X=x)$). As a result, we also emphasized the importance of within-group fairness in the paper.
>  In our setting, we say an algorithm satisfies within-group fairness if, within each protected group, a subgroup involving the most qualified individuals is selected.
>  This is equivalent to saying that if an individual $x$ is more qualified than another individual $x'$, i.e., $P(Y=1|A=a, X=x) > P(Y=1|A=a, X=x')$, then, if $x'$ is selected, $x$ is also selected.
>  In school admission, for example, among both male and female students, students with higher grades (we assume grade is the only common feature for simplicity) should be admitted before students with lower grades. We believe this ``within’’ group fairness is a minimum requirement for any fair algorithm, no matter what kind of group-wise fairness metric is considered.
>
> Q3: Clarification of Equation 3.
>
> Yes, it is a typo, and we have revised it in the paper. Equation 3 indeed considers the case where condition 4.1 is not satisfied (for the case that $A$ is binary). As you mentioned, in theorem 4.2, we only provide a sufficient condition such that the fair Bayes optimal classifier is a GWTR.
> In other words, from Thm 4.2 alone,
> condition 4.1 may be only a technical condition required by the proof, and fair Bayes optimal classifiers could always be GWTRs.
> To clarify this concern, we consider equation 3, which violates condition 4.1, and further prove the negative result in theorem 4.3. Our negative result shows that, unlike demographic parity and equality of opportunity where the fair Bayes optimal classifier is always a GWTR ([3, 4]), the fair Bayes optimal classifier, with respect to predictive parity, is possibly not a GWTR. The interpretation of equation 3 follows theorem 4.3.
>
> Q4: w.r.t. recommended action in Algorithm 1
>
> Indeed, ideally we would like to check if it is possible for the Bayes optimal classifier to achieve PP along with within-group fairness.
> If it is not possible, then we should consider abandoning PP.
> In the previous version, we used the sufficient condition as a proxy that is readily verifiable.
> However, as our thresholding method always generates a GWTR, it will not hurt within-group fairness. As such, we have changed our suggestion from considering another metric to a warning message that, when condition 4.1 is not satisfied, considering predictive parity may lead to large sacrifices in model accuracy.
>
> [1] Dwork, C., Hardt, M., Pitassi, T., Reingold, O., \& Zemel, R. (2012, January). Fairness through awareness. In Proceedings of the 3rd innovations in theoretical computer science conference (pp. 214-226).
>
> [2] H\'{e}bert-Johnson, U., Kim, M., Reingold, O., \& Rothblum, G. (2018, July). Multicalibration: Calibration for the (computationally-identifiable) masses. In International Conference on Machine Learning (pp. 1939-1948). PMLR.
>
> [3] Menon, A. K., \& Williamson, R. C. (2018, January). The cost of fairness in binary classification. In  Conference on Fairness, Accountability and Transparency (pp. 107-118). PMLR.
>
> [4] Zeng, X., Dobriban, E., \& Cheng, G. (2022). Bayes-Optimal Classifiers under Group Fairness. arXiv preprint arXiv:2202.09724.

---

> > ### Comment · Reviewer_m8AG · 2022-08-08
> > **Thank authors for the response**
> >
> > Thank authors for the response. I am hesitant to agree with the claim that " 'within group fairness' is a minimum requirement for any fair algorithm, no matter what kind of group-wise fairness metric is considered ". For example, randomization in the post-processing way can guarantee Equalized Odds (Hardt et al., 2016), and this post-processing does not have any guarantee on "within-group fairness".
> >
> > I understand the fact that group-level fairness alone in general does not guarantee individual-level fairness. However, considering the fact that this paper is focusing on PP notion of fairness (which is not individual-level), I still think the presented discussion on within-group fairness is confusing and (potentially) misleading.

---

> > > ### Author Response · Authors · 2022-08-09
> > > **Response to Further Concerns**
> > >
> > > Thanks a lot for your further comments. We appreciate the time you spent on the paper. We will address your further concerns below.
> > >
> > >
> > > $\\textbf{Concern}$: I am hesitant to agree with the claim that "within group fairness" is a minimum requirement for any fair algorithm, no matter what kind of group-wise fairness metric is considered ". For example, randomization in the post-processing way can guarantee Equalized Odds ([1]), and this post-processing does not have any guarantee on "within-group fairness".
> > >
> > > I understand the fact that group-level fairness alone in general does not guarantee individual-level fairness. However, considering the fact that this paper is focusing on PP notion of fairness (which is not individual-level), I still think the presented discussion on within-group fairness is confusing and (potentially) misleading.
> > >
> > > $\\textbf{Our Response}$: Thanks a lot for your response. We agree that in [1], there is no guarantee of "within-group fairness". As the first paper that introduced equalized odds to fair classification and provided a practical algorithm, [1] is of great theoretical and empirical interest. However, later works, such as [2] and [3], raised concerns, such as that the randomization in [1] delivers the following message:
> > > luck is more important than effort, as long as one achieves a certain level.
> > > We believe this is not desirable for a decision-making system.
> > >
> > > Moreover, under independence-/separation-based measures, the fair Bayes-optimal classifiers are always group-wise thresholding rules. In other words, for demographic parity or equality of opportunity, maximizing accuracy is compatible with within-group fairness. Thus, within-group fairness is not a severe concern as it can be implicitly avoided by maximizing accuracy. However, for PP with certain distribution conditions, the fair Bayes-optimal classifier is not a group-wise thresholding rule. Maximizing accuracy with PP will lead to within-group unfairness. As a result, we believe that we need to emphasize this undesirable consequence (or ethical risk) of PP.
> > >
> > > However, we do not wish to make an overly general claim about within-group unfairness being universal, as this is not our central contribution. We will rephrase this more carefully, to reflect that we simply think it is important here, not universal in general.
> > >
> > > [1] Hardt, M., Price, E., \& Srebro, N. (2016). Equality of opportunity in supervised learning. Advances in neural information processing systems, 29.
> > >
> > > [2] Dwork, C., Hardt, M., Pitassi, T., Reingold, O., \& Zemel, R. (2012, January). Fairness through awareness. In Proceedings of the 3rd innovations in theoretical computer science conference (pp. 214-226).
> > >
> > > [3] H\'{e}bert-Johnson, U., Kim, M., Reingold, O., \& Rothblum, G. (2018, July). Multicalibration: Calibration for the (computationally-identifiable) masses. In International Conference on Machine Learning (pp. 1939-1948). PMLR.

---

### Meta-Review · Area_Chair_PdY5 · 2022-09-02

**Recommendation:** Accept
**Confidence:** Certain

**Metareview:**

The author considers fair Bayes-optimal classifiers under predictive parity. The authors show that under some sufficient conditions all such classifiers are groupwise threshold rules. They also show that htis is not necessarily the case when the condition does not hold. Some empirical results are also provided. The reviewers have made several important suggestions; in particular the authors should provide better context as well as discuss related work using other sufficiency metrics. As the changes are mainly about prior work, I'm inclined to recommend acceptance, but the authors should aboslutely implement all the changes they promised in the responses to the reviews.

**Award:**

No

---

### Decision · Program_Chairs · 2022-09-14

Accept